# Modulation of polypeptide conformation through donor–acceptor transformation of side-chain hydrogen bonding ligands

Ziyuan Song[1], Rachael A. Mansbach [2], Hua He[3], Kuo-Chih Shih [4,5], Ryan Baumgartner[6], Nan Zheng[1], Xiaochu Ba[6], Yinzhao Huang[6], Deepak Mani[1], Yun Liu[7,8], Yao Lin [4,9], Mu-Ping Nieh[4,10], Andrew L. Ferguson [1,11], Lichen Yin[3] & Jianjun Cheng[1]

Synthetic polypeptides have received increasing attention due to their ability to form higher ordered structures similar to proteins. The control over their secondary structures, which enables dynamic conformational changes, is primarily accomplished by tuning the side-chain hydrophobic or ionic interactions. Herein we report a strategy to modulate the conformation of polypeptides utilizing donor–acceptor interactions emanating from side-chain H-bonding ligands. Specifically, 1,2,3-triazole groups, when incorporated onto polypeptide side-chains, serve as both H-bond donors and acceptors at neutral pH and disrupt the α-helical conformation. When protonated, the resulting 1,2,3-triazolium ions lose the ability to act as H-bond acceptors, and the polypeptides regain their α-helical structure. The conformational change of triazole polypeptides in response to the donor-acceptor pattern was conclusively demonstrated using both experimental-based and simulation-based methods. We further showed the utility of this transition by designing smart, cell-penetrating polymers that undergo acid-activated endosomal escape in living cells.

[1] Department of Materials Science and Engineering, University of Illinois at Urbana–Champaign, Urbana, Illinois 61801, USA. [2] Department of Physics, University of Illinois at Urbana–Champaign, Urbana, Illinois 61801, USA. [3] Jiangsu Key Laboratory for Carbon-Based Functional Materials & Devices, Institute of Functional Nano & Soft Materials (FUNSOM), Soochow University, Suzhou 215123, China. [4] Polymer Program, Institute of Materials Science, University of Connecticut, Storrs, Connecticut 06269, USA. [5] Department of Agricultural Chemistry, National Taiwan University, Taipei 10617, Taiwan. [6] Department of Chemistry, University of Illinois at Urbana–Champaign, Urbana, Illinois 61801, USA. [7] Center for Neutron Research, National Institute of Standards and Technology, Gaithersburg, Maryland 20899, USA. [8] Department of Chemical and Biomolecular Engineering, University of Delaware, Newark 19716, USA. [9] Department of Chemistry, University of Connecticut, Storrs, Connecticut 06269, USA. [10] Department of Chemical and Biomolecular Engineering, University of Connecticut, Storrs, Connecticut 06269, USA. [11] Department of Chemical and Biomolecular Engineering, University of Illinois at Urbana–Champaign, Urbana, Illinois 61801, USA. Correspondence and requests for materials should be addressed to A.L.F. (email: alf@illinois.edu) or to L.Y. (email: lcyin@suda.edu.cn) or to J.C. (email: jianjunc@illinois.edu)

Hydrogen bonding (H-bonding) interactions are one of the most important non-covalent molecular forces in biology, chemistry, and materials science[1–4]. Compared to other molecular forces including hydrophobic and electrostatic interactions, the alignment of the donor–acceptor pair constituting a H-bond restricts the geometry of the interaction. Furthermore, the pattern of H-bond donors and acceptors within molecules capable participating in multiple H-bonds provides specificity by ensuring H-bonding interactions between complementary molecules[5]. These two unique properties of H-bonding interactions are elegantly utilized in nature to construct the precise three-dimensional structures of nucleic acids and proteins[6, 7]. For instance, α-helices and β-sheets are formed and stabilized through H-bonds between backbone carbonyls and N–H groups, where all H-bond donors and acceptors are paired with nearly straight geometry[8, 9]. The added benefit of H-bonds is that they are relatively weak, enabling macromolecular structures to undergo dynamic remodeling; a trait that is widely utilized in reaction pathways and processes essential for life, such as the transcription of DNA and the conformational changes of proteins[7, 10].

Inspired by nature, several biomimetic materials have been developed whose higher ordered structures are also constructed and maintained through H-bonding interactions, including peptidomimetic polymers[11, 12], foldamers[13, 14], and supramolecular polymers[15]. Among these materials, synthetic polypeptides have received increasing attention as protein mimics due to their ability to form important secondary structures such as α-helices. The ability to synthetically introduce unnatural components into polypeptides[16] has widened the scope of these materials and provides new insights into novel biomaterials design[11, 17]. Previous work in relation to these unnatural polypeptides has revealed the importance of hydrophobic[18–20] and Coulombic[18, 21] interactions in stabilizing or destabilizing the α-helical conformation. The understanding of these interaction has enabled the synthesis of several polypeptide materials that are able to

respond to changes in their environment and undergo helix-coil transitions[20–24]. While hydrophobic and ionic interactions interfere with backbone H-bonds of polypeptides indirectly, it remains challenging to directly manipulate backbone H-bonds of polypeptides. The direct manipulation of H-bonds provides a more responsive transition behavior and has been demonstrated in materials such as foldamers, where the addition or removal of a single H-bond at the chain end is able to completely alter the conformation of an oligopeptide[14, 25]. Inspired by these materials, we were curious whether similar competing interactions introduced within a polypeptide side-chain would also provide a sensitive response to environmental changes, drastically altering its overall structure.

Here, we report an approach to modulate the secondary structure of polypeptides through the transformation of donor-acceptor H-bonding ligands incorporated on the side-chains. Compared to previously reported systems with hydrophobic and ionic interactions, this strategy is advantageous due to the versatile design of H-bonding ligands, the precise control of donor-acceptor patterns, and the ease of altering the H-bonding pattern under mild conditions. The change in conformation of these polypeptides in response to the donor–acceptor identity of the side-chain is confirmed through circular dichroism (CD) spectroscopy, molecular dynamics (MD) simulations, and small angle neutron scattering (SANS). We further demonstrate that the change in secondary structure of these polypeptides can be utilized to design cell-penetrating polypeptides with trigger-activated membrane penetration capability. The work provides insights into the control over the higher ordered structures that are held by H-bonding interactions, which can be further utilized in the design of functional materials.

## Results

**Impact of polypeptide side-chain ester-to-amide modification.** The impact of side-chain H-bonding pattern on the polypeptide conformation is illustrated in Fig. 1. In our previous work on α-helical cationic polypeptides, we demonstrated that poly (L-glutamic acid) derivatives with elongated hydrophobic ester side-chains (**PE**, Fig. 2a) are able to maintain a stable helical conformation over a broad range of pH values[18]. Recently, however, we observed that exchanging the side-chain ester group with an amide (**PE** to **PA**, Fig. 2a) led to complete disruption of the α-helix (Fig. 2d). This change in conformation is unusual as it does not give rise to ionic interactions that are known to inhibit the formation of the α-helix, nor does it provide significantly increased steric interactions. The substitution, however, does greatly increase the polarity of the carbonyl and adds an amidic H-bond donor (Fig. 2b)[26]. Given the proximity and structural similarity of the side-chain amides to those contained within the polypeptide backbone, it is possible that this side-chain amide participates in H-bonding interactions with the backbone and interrupts the α-helical structure. The ester in **PE** has a lower propensity to participate in these interactions as its H-bonding pattern consists of only a H-bond acceptor, the carbonyl. The term H-bonding pattern here refers to the specific role of a functional group in the H-bonding interaction. For example, the ester, containing only a carbonyl group capable of interacting in a H-bonding interactions, is classified as having a unitary H-bonding (UHB) pattern with either a H-bond donor or a H-bond acceptor. Similarly, the amide group in **PA** contains both a H-bond donor and an acceptor and is classified as being a binary H-bonding (BHB) pattern. Based on the data, it is our contention that the BHB groups, such as the side-chain amides in **PA**, interfere with the H-bonds of the polypeptide backbone, and thereby disrupt the α-helix. On the contrary, the α-helical

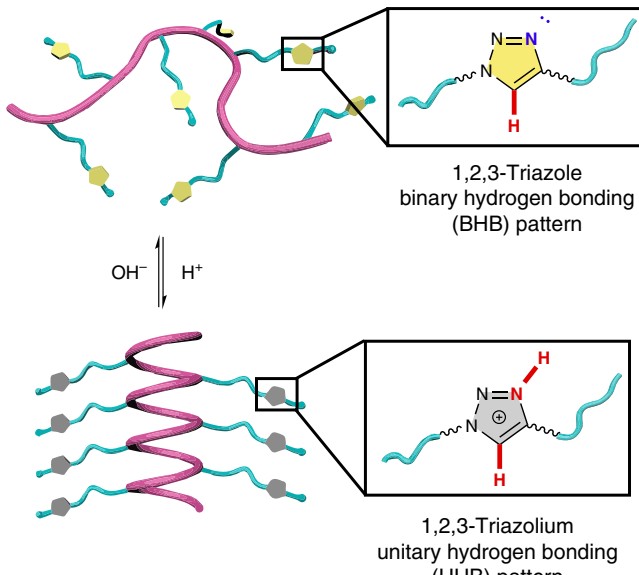

**Fig. 1** Illustration showing the regulation of polypeptide conformation by H-bonding. The secondary structure of polypeptides were regulated through donor-acceptor transformation of side-chain H-bonding ligands. The protonation of 1,2,3-triazole (binary H-bonding pattern) to 1,2,3-triazolium (unitary H-bonding pattern) induced the coil-to-helix transition of the polypeptides. H-bond donors and acceptors are highlighted in *red* and *blue*, respectively

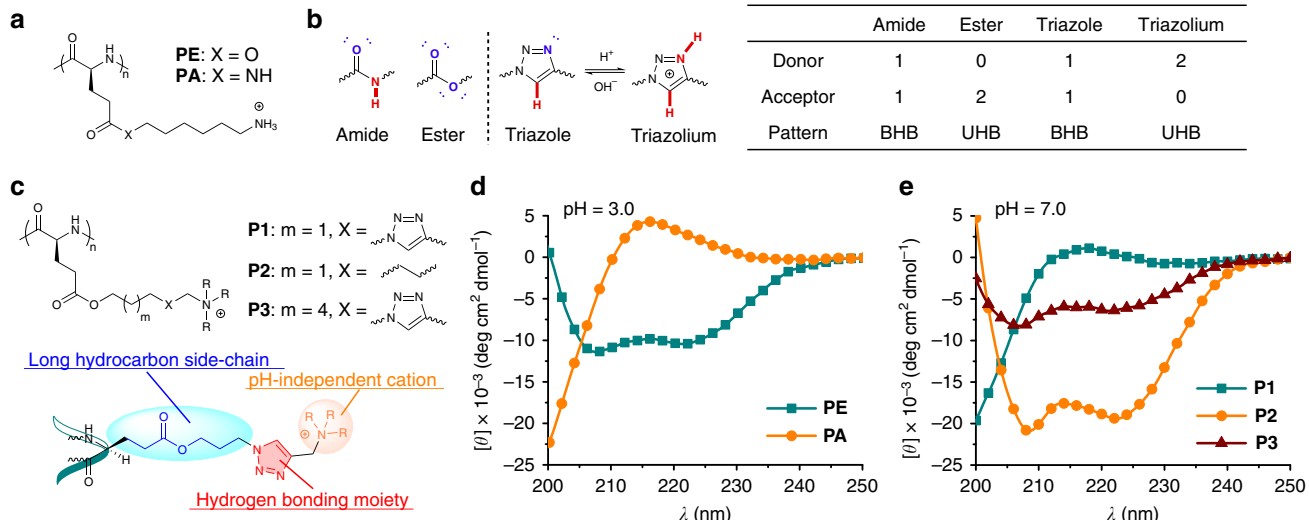

**Fig. 2** Side-chain triazole groups disrupt the backbone α-helical conformation. **a** Chemical structures of **PE** and **PA**. **b** H-bonding pattern analysis of amide, ester, 1,2,3-triazole, and 1,2,3-triazolium. H-bond donors and acceptors are highlighted in *red* and *blue*, respectively. **c** Chemical structures of **P1-P3**. The molecular design of triazole polypeptides is highlighted, where each component is essential for the study. **d**, **e** CD spectra of polypeptides. **PE** and **PA** were analyzed at pH 3.0 (**d**), and **P1-P3** were analyzed at pH 7.0 (**e**)

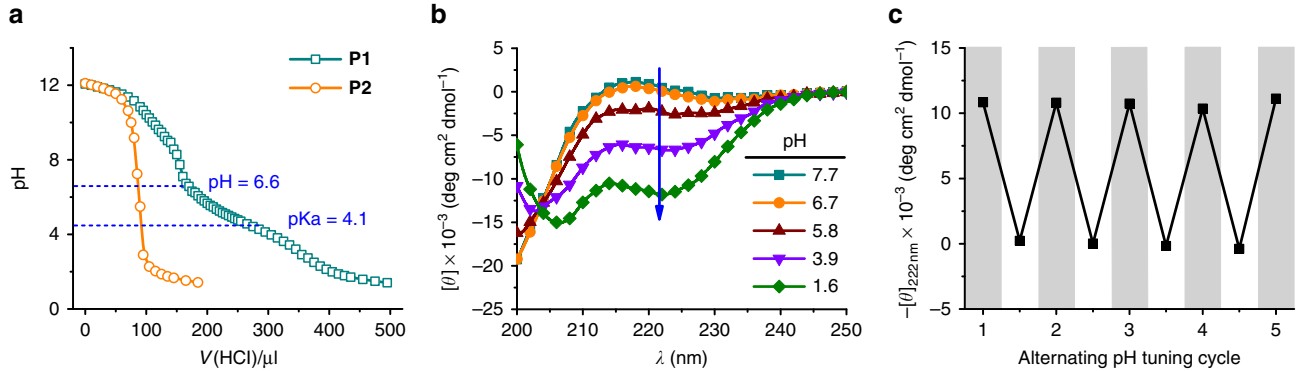

**Fig. 3** Coil-to-helix transition induced by protonation of side-chain 1,2,3-triazole. **a** The pH titration curve of **P1** with side-chain triazoles and **P2** without side-chain triazoles. **b** CD spectra of **P1** upon stepwise addition of concentrated HCl. **c** The mean residue molar ellipticity of **P1** at 222 nm ($-[\theta]_{222\,nm}$) after alternating addition of HCl and NaOH over five cycles. At pH ~ 2.8 (*light grey background*), **P1** adopts an α-helical conformation with a positive $-[\theta]_{222\,nm}$ value; at pH ~ 9.5 (*white background*), the value of $-[\theta]_{222\,nm}$ is approximately zero indicating a random coil structure

conformation is stable for **PE** and its derivatives[18, 27–29], as their side-chains display only H-bond accepting capability (UHB pattern).

**Side-chain triazoles disrupt the α-helical structure**. To examine the above hypothesis, we were interested in incorporating other amide bond isosteres onto the polypeptide side-chains. These moieties bearing a BHB pattern are expected to disrupt the backbone helix in a manner similar to **PA**. One isostere of the peptide bond we were particularly interested in studying was the 1,2,3-triazole group[30], whose C5 hydrogen functions as the H-bond donor and N3 acts as the acceptor (Fig. 2b). In analogy to the amide groups, both H-bond donating and accepting interactions mutually strengthen each other[30]. Accessing these groups is also easily accomplished as triazoles can be synthesized via Huisgen click chemistry[31], providing access to several polypeptide structures in order to elucidate important structure-property relationships. Most importantly, we were interested in studying how the transformation between BHB and UHB groups in-situ would affect the polypeptide conformation. For this reason, the 1,2,3-triazole is advantageous, as protonation at the N3 site

inhibits the ability of this group to participate as a H-bond acceptor. The resulting 1,2,3-triazolium cation, instead, possesses two weaker H-bond donors (UHB pattern, Fig. 2b)[30]. According to our analysis on **PE** and **PA**, it is expected that protonation of the triazole will thus lead to recovery of the H-bond network of the α-helix by altering the side-chain H-bonding pattern from BHB to UHB.

We therefore developed a synthetic strategy to access a model cationic polypeptide **P1** (50 mer) bearing 1,2,3-triazole side-chains through the ring-opening polymerization of chlorine-based *N*-carboxyanhydride monomers and subsequent functionalization of side-chains to azides. A Huisgen click reaction with functional alkynes then provided the final functionalized polypeptides containing 1,2,3-triazole[27]. The deliberate use of charged quaternary trimethylammonium groups located at the side-chain terminus provides pH-independent water solubility of **P1**, and the long hydrocarbon side-chains ensure adequate separation of the ammonium groups from the peptide backbone which would otherwise disrupt the α-helix (Fig. 2c)[18]. The adequate separation was confirmed with a control polypeptide, **P2**, which lacks the triazole linkages and adopts a typical α-helical conformation at neutral pH (Figs. 2c, e). Under identical

conditions, however, the incorporation of 1,2,3-triazoles onto the side-chains (**P1**) led to a change in the conformation of the peptide backbone to a random coil (Fig. 2e). This result reveals the disruptive effect of the triazole on the α-helix in direct analogy with the amide-based polypeptide **PA**. To further support the central role of the triazole group, we extended the distance of the triazole from the peptide backbone in attempts to attenuate the effect of the triazole. Unlike **P1**, which adopts a random coil structure due to inhibitory action of the triazole, the polypeptide **P3** displayed an α-helical conformation (Figs. 2c, e) due to both enhanced side-chain hydrophobic packing and sequestration of the disruptive triazole moieties from the backbone. In order to rule out concomitant extension of the ammonium groups from the backbone in **P3**, we synthesized polymers **P4** and **P5** that contained extended linkers between the triazole and the ammonium groups. These polypeptides, like **P1**, displayed a random coil structure (Supplementary Fig. 1), further confirming that the disruptive effect originates from the triazole groups rather than the ammoniums groups.

**Protonation of triazole induces coil-to-helix transition.** In order to confirm the basic nature of 1,2,3-triazole and determine the pH range under which it will accept a proton, a pH titration of **P1** was carried out in aqueous solution, revealing a pKa of ~4.1 (Fig. 3a). While higher than the reported value of the small molecule 1-methyl-1,2,3-triazole (pKa ~ 1.3[32]), the micro-environment of the polymer often moderates pKa values (e.g., in the case of poly(L-glutamic acid)[33]). As a comparison, **P2**, which

lacks triazole groups, did not reveal a buffering effect (Fig. 3a). Protonation of the triazole was also verified using [1]H NMR through a significant downfield shift of the methylene protons adjacent to triazole C1 as the pH was lowered (Supplementary Fig. 2).

The protonation of triazole at weakly acidic pH leads to changes in its H-bonding pattern (Fig. 2b), which should subsequently induce changes in the polypeptide conformation according to our hypothesis. To test this, we analyzed the CD spectra of **P1** at varying pH values. While **P1** remained as a random coil at neutral and basic pH, it began to adopt an α-helical conformation as the pH was lowered (Fig. 3b). The starting point of the shift in conformation observed by CD coincided with the onset of the buffering effect of the titration curve (pH ~ 6.6), indicating the correlation between the inhibition of disruptive effect on α-helix and the protonation of side-chain triazole. The appearance of an isodichroic point at 203 nm is consistent with a two-state helix-coil transition and suggests the absence of other secondary structure during the transition[34]. As a comparison, control polypeptide **P2** without side-chain triazole linkage remained as an α-helix with negligible ellipticity changes upon decreasing the pH (Supplementary Fig. 3). The conformational change was also verified by [1]H NMR, where the sharp peaks for all backbone protons of **P1** became weaker and broader upon decreasing the pH, mainly due to side-chain shielding upon polypeptide folding (Supplementary Fig. 2)[35]. All these results collectively indicate that protonation of the triazole into a triazolium indeed results in removal of the inhibitory BHB pattern and recovery of the helix in situ. This pH

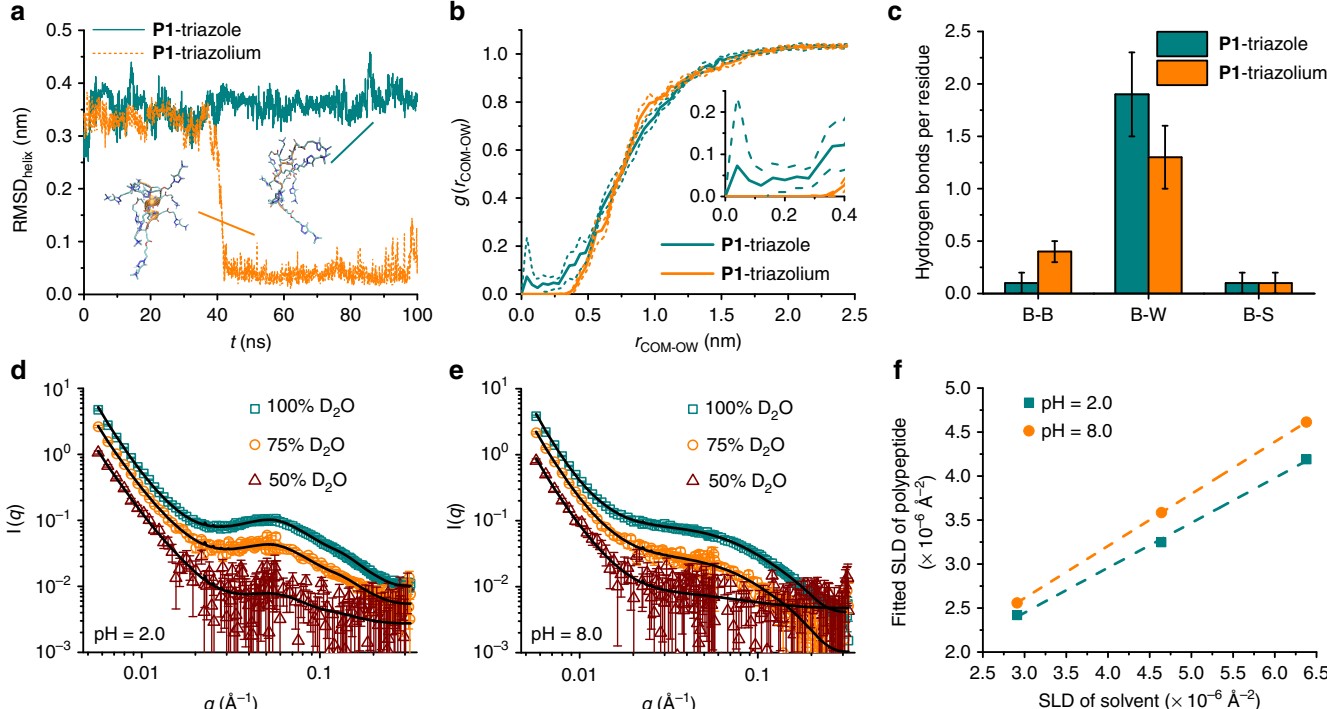

**Fig. 4** Molecular dynamics simulations and small angle neutron scattering tests of **P1**. **a** Time trace of the root mean squared deviation of the backbone Cα atoms from an ideal α-helix ($RMSD_{helix}$). Representative snapshots of each polypeptide were visualized in VMD[52], where water molecules have been removed for clarity of viewing. **b** Radial distribution function between the polypeptide backbone center of mass and the water solvent O atoms, $g(r_{COM-OW})$, for **P1**-triazole and **P1**-triazolium over the final 50 ns of the simulation employing a bin size of 0.04 nm. The *thin dotted lines* flanking each curve denote standard errors estimated by block averaging over five 10 ns blocks. The inset shows an enlarged view of the water depletion zone. **c** Histogram of the number of H-bonds per residue within the polypeptide backbone (B–B), between the backbone and water (B–W), and between the backbone and side-chain triazole/triazolium (B–S). Error bars are standard deviations about the mean of the distributions of H-bonds per residue. Each distribution contains 1000 data points. **d**, **e** Scattering patterns of **P1** in different $D_2O/H_2O$ solvents at pH 2.0 (**d**) and pH 8.0 (**e**). Error bars represent standard deviations derived from neutron counts. **f** Linear regression of the fitted scattering length density (SLD) of the polypeptide and the SLD of the solvent

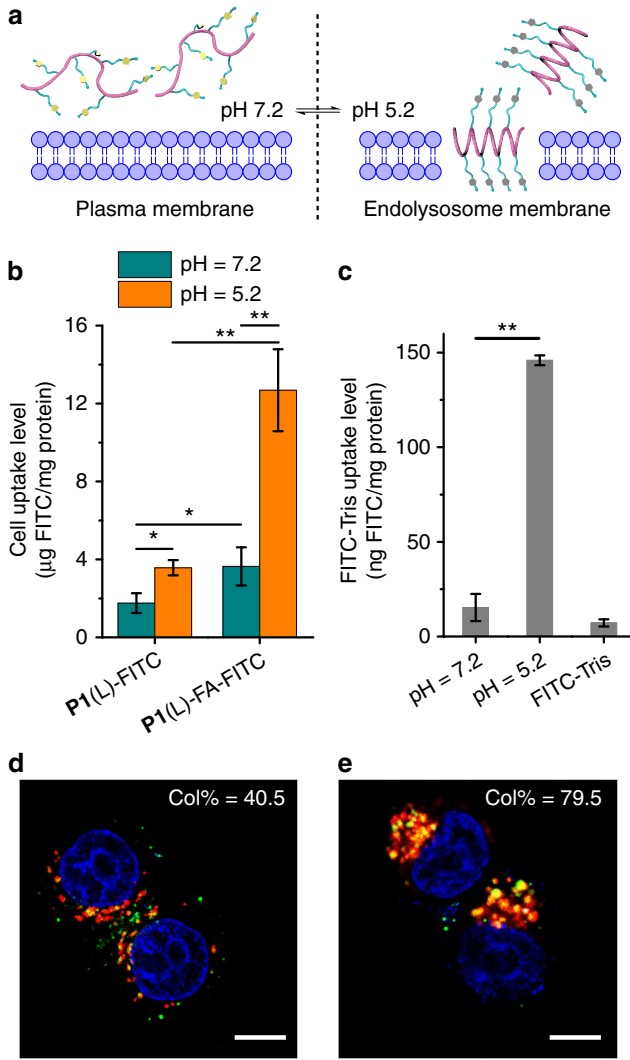

**Fig. 5** Triazole polypeptides mediate cancer cell-targeted internalization and acid-induced endosomal escape. **a** Scheme showing pH-dependent membrane penetration of **P1**. **b** Cell uptake of **P1**(L)-FITC and folic acid (FA)-modified polypeptide **P1**(L)-FA-FITC in HeLa cells at pH 7.2 and 5.2. Results represent the means ± s.e.m. of three replicates. **c** Uptake level of FITC-Tris in HeLa cells after co-incubation with **P1** at pH 7.2 and 5.2. Results represent the means ± s.e.m.of three replicates. **d, e** Merged CLSM images of HeLa cells following incubation with **P1**(L)-FA-FITC (**d**) or **P1**(DL)-FA-FITC (**e**) at 37 °C for 4 h. Cell nuclei were stained with Hoechst 33258 (*blue*) and endosomes/lysosomes were stained with Lysotracker Red (*red*). Scale bar represents 10 μm. Col% represents the colocalization ratio of **P1**(L)-FA-FITC or **P1**(DL)-FA-FITC (*green*) with Lysotracker Red (*red*) (*n* = 50)

induced conformational transition is also highly reversible as expected. After five cycles from pH ~9.5 to ~2.8 and back, the α-helix demonstrated complete recovery of ellipticity at the low pH condition (Fig. 3c). The reversible nature of this transition additionally excludes the possibility of irreversible chemical changes, such as the hydrolysis of side-chain esters or the isomerization of the backbone α-carbon.

**MD simulation of triazole polypeptides**. To confirm the impact of the side-chain H-bonding pattern on the conformation of the polypeptides, we performed all-atom MD simulations to probe the conformation of molecules. This technique has already been widely used to study the folding process of proteins[36]. **P1**-triazole

(no triazoles protonated, BHB pattern) and **P1**-triazolium (all triazoles protonated, UHB pattern) were selected to represent the triazole polypeptide at neutral and acidic pH, respectively (Supplementary Fig. 4). Polypeptide chains were modeled using the GROMOS 54A7 force field[37] and denatured into unstructured random coils by applying an artificial stretching potential between the terminal Cα atoms. Each polypeptide was then placed in a cubic box with periodic boundary conditions and solvated by simple point charge (SPC) water molecules[38] at 298 K and 1 bar. During the 100 ns simulation, **P1**-triazole remained as a random coil, whereas **P1**-triazolium spontaneously folded into an α-helix after ~40 ns, and remained in that state for the duration of the run, indicating a stable α-helix formation (Fig. 4a, Supplementary Movies 1 and 2). Excluding the two terminal residues to reduce chain-end effects, **P1**-triazolium adopts a nearly ideal α-helix with a helical radius $r_{helix} = (0.233 \pm 0.004)$ nm and a twist angle $\gamma_{helix} = (98 \pm 1)°$ in excellent agreement to that of an ideal α-helix ($r_{helix}^{ideal} = 0.23$ nm, $\gamma_{helix}^{ideal} = 100°$) (Supplementary Fig. 4). The average root mean squared deviation of the backbone Cα atoms from an ideal α-helix ($RMSD_{helix}$) is $(0.04 \pm 0.02)$ nm (Fig. 4a). In order to compare the results of the simulation with experimental data, we synthesized a short 10 mer of **P1** and analyzed it by CD spectroscopy under neutral and acidic conditions (Supplementary Fig. 5). The experimental mean residue molar ellipticity values ($[\theta]_{222\,nm}$) are in excellent agreement with the prediction from simulation trajectories using DichroCalc[39]. At neutral pH, the 10 mer of **P1** adopts a random coil conformation and possesses a $[\theta]_{222\,nm}$ value of $-0.3 \times 10^3$ deg cm$^2$ dmol$^{-1}$, agreeing with that predicted from MD simulation $(-1.8 \pm 2.8) \times 10^3$ deg cm$^2$ dmol$^{-1}$. Under acidic conditions where **P1** adopts a helical conformation, the measured ellipticity shifts to $-7.4 \times 10^3$ deg cm$^2$ dmol$^{-1}$ and is again in agreement with the value of $(-9.2 \pm 1.5) \times 10^3$ deg cm$^2$ dmol$^{-1}$ predicted by simulation. The slightly reduced ellipticity value of **P1** at pH 1.8 compared to the simulated value likely stems from the incomplete protonation of the triazole side-chains experimentally.

The trajectories from MD simulations allow us to further understand the detailed structure of **P1** at the atomic level. Consistent with its random coil conformation, the center of mass of **P1**-triazole is accessible to water molecules (Fig. 4b), indicating the complete hydration of backbone peptide bonds. More than 90% of the backbone amides form H-bonds with the solvent water in **P1**-triazole (Fig. 4c), suggesting the influx of water helps to stabilize the free backbone carbonyl/N–H groups. The key role of backbone hydration in enabling the disruption of the α-helix is also supported by a non-ionic control triazole polypeptide, **P6**, which adopts a stable α-helical conformation in water-free organic solvent (Supplementary Fig. 6). As a comparison with **P1**-triazole, a ~0.4 nm water depletion zone is present around the center of mass of the helical **P1**-triazolium due to the compact packing of backbone atoms (Fig. 4b). Compared with **P1**-triazole, the analysis of the H-bonding partners of **P1**-triazolium reveals an increase in the number of H-bonds between peptide backbones, accompanied by a reduction in the number of backbone-water H-bonds, agreeing with the transition to an α-helical conformation (Fig. 4c). The relatively low degree of backbone peptide-peptide H-bonding even at α-helical conformation (0.4 per residue) is attributed to the dangling peptide bond at the chain termini (~four dangling carbonyl/N–H groups at each end, the analysis of the H-bonding partners eliminating the chain-end effect is shown in Supplementary Fig. 7).

**Small angle neutron scattering of triazole polypeptides**. The MD simulations provide useful information concerning the polypeptide conformation and the hydration of backbones, but

are limited to the study of short peptides (10 mer) under ideal conditions (100% protonation of side-chains for **P1**-triazolium). In order to further characterize the actual structure adopted by longer (50 mer), incompletely protonated polypeptides **P1**, SANS was conducted at different pH values. SANS is a useful technique to probe the structural information of biological polymers, including the size, shape, and water content[40]. Three solvent conditions with varying $D_2O/H_2O$ fractions ($D_2O = 100$, 75, and 50%, v/v) were used to dissolve **P1** (50 mer) at both acidic (pH = 2.0) and basic conditions (pH = 8.0). The scattering curves of **P1** at both pH values indicate an initial $q^{-3.5} \sim q^{-3.7}$ decay (for $q = 0.006 \sim 0.017 \text{ Å}^{-1}$), presumably attributed to the surface scattering of some large aggregates. Afterwards (i.e., $q > 0.017 \text{ Å}^{-1}$) the SANS data can be described by a cylindrical form factor for both samples. Additionally, a Hayter-Penfold structure factor[41, 42] accounting for the Coulombic repulsion between polymer chains is included for both conditions. (Figs. 4d, e). The random coiled polymer at basic pH was also fit with a cylindrical form factor to make a good comparison with the helical polymer at acidic pH.

The morphology and neutron scattering length density (SLD) of the **P1** cylinder were then obtained through the simultaneous fitting process on the samples with different $D_2O/H_2O$ compositions. The optimized fitting parameters for **P1** under both acidic and basic conditions resulted in similar radii (~1.3 nm). The cylindrical length for **P1** under acidic conditions (~7.5 nm) is slightly longer than that under basic conditions (~6.2 nm), indicating a more extended length for the helical conformation compared to the coil structure. The volume fraction of water, based on the SLDs under different contrast conditions, was determined to be 51 and 59% for pH = 2.0 and pH = 8.0 samples, respectively (from the slopes of the lines in Fig. 4f; see Supplementary Methods and Supplementary Eq. 1). The lower fraction of water under acidic condition agrees with the observation of a water depletion zone (Fig. 4c) in **P1**-triazolium (10 mer) from simulation trajectories, confirming the α-helical conformation of **P1** at pH 2.0 even the side-chain triazoles are not 100% protonated.

**Acid-activated endolysosomal membrane penetration**. We have previously demonstrated that helical structure dramatically contributes to the membrane activity of polypeptides[24, 27–29]. Considering the pH-responsive conformation transition of **P1**, we reasoned that the triazole polypeptide may serve as a promising cell penetrating peptide (CPP) mimic to impart selective membrane penetration in response to different biological pH values. CPPs are widely used as molecular transporters because of their potent capability to penetrate the cellular and endosomal membranes, featuring efficient cellular uptake and endosomal escape of the cargo molecules[43]. However, CPPs can penetrate the plasma membranes of all cell types, which often leads to undesired side effects in non-target cells. According to our previous studies on the helicity-associated cell penetrating ability of polypeptides, the triazole polypeptides are promising candidates in enabling strong endosomal disruption in its helical state at the acidic endosomal pH (4.5~5.5), while affording minimal cellular internalization in its coiled state at the neutral extracellular pH (6.8~7.4) (Fig. 5a)[44]. When coupled with cell-specific targeting ligands, the coiled triazole polypeptide can be selectively internalized into target cells, and subsequently mediate effective endosomal/lysosomal escape upon conformational transition to the membrane-active, helical state in the endosome/lysosomes. To provide evidence for this hypothesis, we first labeled the polypeptides with fluorescein and evaluated their uptake levels in HeLa cells at pH 5.2 and 7.2. As shown in Fig. 5b and Supplementary Fig. 11, **P1**(L)-FITC afforded a 3-fold higher

uptake level at pH 5.2 than at pH 7.2, indicating acid-activated membrane penetrating capability. Consistently, **P1**-mediated uptake of FITC-Tris, a membrane-impermeable biomarker, was ~10-fold higher at pH 5.2 than at 7.2, indicating that the acid-activated helical triazole polypeptides mediate pore formation on membranes (Fig. 5c)[24]. Considering its low membrane activity at neutral pH, we further modified the polypeptide with folic acid (FA), a cancer cell targeting ligand, to promote selective internalization into cancer cells via receptor-mediated endocytosis (Supplementary Figs 9 and 10). The **P1**(L)-FA-FITC indeed showed significantly higher uptake level in HeLa cells at pH 7.2 than **P1**(L)-FITC, and it was further augmented by 3.5 fold when the pH was decreased to 5.2 (Fig. 5b). Consistent with FA-mediated cancer cell targeting and the acid-activated membrane activity, **P1**(L)-FA-FITC showed appreciable cytoplasmic distribution in HeLa cells, and notable separation between green fluorescence (**P1**(L)-FA-FITC) and red fluorescence (Lysotracker Red-stained endolysosomes) was observed using confocal laser scanning microscopy (CLSM), which provided strong evidence that the triazole polypeptide can selectively traverse into cancer cells via FA-mediated targeting effect followed by potent endolysosomal disruption upon pH-triggered coil-to-helix transition (Fig. 5d, Supplementary Fig. 12 and Supplementary Fig. 13). In comparison, the racemic polypeptide analog **P1**(DL)-FITC and **P1**(DL)-FA-FITC, adopting random coil conformation at both neutral and acidic pH (Supplementary Fig. 9), exhibited similarly low cell penetration levels at both pH values (Supplementary Fig. 11), and thus showed weak capability in inducing endosomal escape (Fig. 5e, Supplementary Figs 12 and 13).

## Discussion

The modulation of the polypeptide backbone conformation via the side-chain H-bonding pattern is reminiscent of the widely used protein denaturant urea, which possesses H-bond donor (amide hydrogens) and acceptor (carbonyl oxygen) groups that endow it with a BHB pattern[45, 46]. Two principal theories have been proposed for its mechanism of action: the direct mechanism wherein urea binds to and stabilizes the denatured state, and the indirect mechanism wherein urea modifies the water H-bonding network to suppress the hydrophobic effect and favor the exposure of buried hydrophobic residues[45]. Despite a plethora of primarily simulation but also some experimental studies, the precise molecular mechanism remains poorly understood[45, 46]. Drawing an analogy with this body of work, we suggest that the side-chain BHB moieties may disrupt the backbone α-helical conformation through one or both of these direct and indirect pathways, and that resolution of the precise molecular mechanism will prove to be rather challenging.

In summary, we have demonstrated that side-chain moieties capable of undergoing H-bond donor-acceptor interactions can be utilized to alter the conformation of the polypeptide. The polypeptides bearing BHB pattern based side-chain groups adopted random coil conformations, whereas the backbone of the polypeptides folded into an α-helix when the side-chain H-bonding pattern became UHB. Specifically, we showed that 1,2,3-triazole containing polypeptides can utilize this mechanism to undergo a reversible conformational change between the α-helical and random-coiled state in response to the change in the H-bonding pattern of side-chain ligands. Through the molecular design of cancer targeting and acid-activated membrane penetration, we were able to achieve the selective uptake into cancer cells while minimizing the entry into normal cells, and subsequently potentiating the escape from endolysosomes to allow the therapeutic efficacy of various payloads. The facile

synthesis of triazole polypeptides using click chemistry allows the incorporation of a variety of functional groups into the material design, making this system a valuable platform for basic mechanistic studies on polypeptide conformation as well as for the design of new responsive materials.

## Methods

**Materials and methods**. All chemicals were purchased from Sigma-Aldrich (St. Louis, MO, USA) and used as received unless otherwise specified. HeLa cells (human cervix adenocarcinoma, folate receptor (FR)-positive) and NIH-3T3 cells (mouse embryonic fibroblast, FR-negative) were purchased from the American Type Culture Collection (Rockville, MD, USA) and cultured in Dulbecco's modified eagle medium (DMEM) supplemented with 10% fetal bovine serum. Detailed instrument setup and polypeptide synthesis can be found in Supplementary Methods.

**CD spectroscopy**. Polypeptides were dissolved in DI water at a concentration of 0.06 or 0.6 mg/ml. The pH value of the solution was adjusted by adding a specific volume of concentrated HCl or NaOH, and the solution pH was measured by pH meter (Oakton Instruments, Vernon Hills, IL, USA). After the pH was adjusted to the desired value, the polypeptide solution was transferred to a quartz cuvette (pathlength = 1 or 10 mm) for CD tests.

**Simulation methods**. We conducted all-atom MD simulations of **P1**-triazole and **P1**-triazolium using the GROMACS 4.6 simulation suite[47]. Initial polypeptide configurations were produced using the Bax Group PDB Utility Server (http://spin. niddk.nih.gov/bax/nmrserver/pdbutil) to construct helical polypeptide backbones, the Automated Topology Builder (ATB) server (http://compbio.biosci.uq.edu.au/atb/)[48–50] to generate side-chain coordinates, and an in-house code to graft the side-chains to the backbone. Polypeptides were modeled using the GROMOS 54A7 force field[37] augmented with the bonded and non-bonded interactions for the synthetic side-chains computed using the ATB server[48–50]. Polypeptides were prepared as zwitterions. Partial charges for the backbone atoms were assigned from the GROMOS 54A7 force field, and those on the side-chains assigned from quantum mechanical predictions using GAMESS-US at a B3LYP/6-31G* level of theory[51]. The high-pH triazole polypeptide possesses a net charge of +10 due to the positively charged terminal trimethylammonium groups on each side-chain. The low-pH triazolium polypeptide carries a net charge of +20 due to additional protonation of each N3 in the triazole ring. Full details of the simulation protocol and analyses are provided in Supplementary Methods.

**Small angle neutron scattering**. SANS experiments were conducted at the NGB 30mSANS located at the National Institute of Science and Technology (NIST) Center for Neutron Research (NCNR, Gaithersburg, MD, USA). The SANS data were collected at two different sample-to-detector distances (7 and 4 m) using 6 Å wavelength neutrons. The two instrument configurations covered a $q$ range from 0.0057–0.32 Å$^{-1}$. Detailed sample preparation and data analysis can be found in Supplementary Methods.

**Confocal laser scanning microscopy**. HeLa and NIH-3T3 cells were seeded on coverslips in 24-well plates at $1.5 \times 10^4$ cells/well and were incubated for 24 h before treatment with various fluorescein-labeled polypeptides at 40 μg polypeptide/well for 4 or 8 h. Cells were washed three times with PBS, stained with Hoechst 33258 (5 μg/ml) and Lysotracker Red (200 nM) before CLSM observation. Colocalization ratio was determined using the LAS AF software (Heidelberg, Germany).

**pH-dependent cell uptake of triazole polypeptides**. HeLa cells were seeded on 96-well plates at $1 \times 10^4$ cells per well and cultured for 24 h before reaching confluence. The medium was changed to opti-MEM (pH 7.2 and 5.2) and fluorescein-labeled polypeptides were added at 2 μg per well. After incubation at 37 °C for 4 h, cells were washed three times with PBS before being lysed with the RIPA lysis buffer (100 μl per well). The content of fluorescein-labeled polypeptides in the cell lysate was quantified by spectrofluorimetry ($\lambda_{ex} = 494$ nm, $\lambda_{em} = 518$ nm) and the protein content was measured by the BCA kit. Uptake level was expressed as μg polypeptide associated with 1 mg cellular protein.

**Statistical analysis**. Statistical analysis was performed using the student's $t$-test and differences were judged to be significant at *$p < 0.05$ and highly significant at **$p < 0.01$.

**Data availability**. The data that support the findings of this study are available within the paper and its Supplementary Information files and available from the corresponding author upon request.

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

## Acknowledgements

This work is supported by NSF-CHE (1308485), NSF-CBET (1510468), and National Natural Science Foundation of China (51403145 and 51573123). Access to NGB30mSANS was provided by the Center for High Resolution Neutron Scattering, a partnership between the National Institute of Standards and Technology and the National Science Foundation under Agreement No. DMR-1508249. The statements, findings, conclusions, and recommendations are those of the authors and do not necessarily reflect the view of NIST or the U.S. Department of Commerce. Identification of a commercial product does not imply recommendation or endorsement by NIST, nor does it imply that the product is necessarily the best for the stated purpose.

## Author contributions

Z.S., M.-P.N., A.L.F., L.Y., and J.C. conceived and designed the experiments. Z.S., R.B., X.B., and Y.H. performed the polypeptide synthesis and characterization. Z.S., X.B., and Y.H. performed the secondary structure studies of polypeptides. R.A.M., D.M., and A.L.F. performed the molecular dynamics simulations. H.H., N.Z., and L.Y. performed the cell studies. K.-C.S., Y.L., and M.-P.N. performed the small-angle neutron scattering experiments. Z.S., R.A.M., K.-C.S., R.B., Y.L., M.-P.N., A.L.F., L.Y., and J.C. analyzed the data and prepared the manuscript with contributions from all authors.

## Additional information

**Competing interests:** The authors declare no competing financial interests.

