## [Peer Review File · Nature Communications]

Reviewers' comments:

Reviewer #1 (Remarks to the Author):

This is a clever and interesting paper that deserves publication in Nature Materials. The synthetic chemistry is original and effective and the end result of switchable hydrogen bonding can have many useful follow-ons. I think the paper is very clear, except on one point. That is in the application to cell-penetration. I am completely convinced of the cell penetration observed in Figure 5. I am much less convinced of the evidence for endosomal escape. All of the fluorescence in Figure 5 appears punctate to me, indicating compartmentalization in endosomes. If this point is clarified, I am very enthusiastic about this paper.

Reviewer #2 (Remarks to the Author):

The authors have developed a new strategy to modulate the conformation of polypeptides via proton acceptor-donor interactions arising from the side-chain H-bonding ligands. They have clearly demonstrated that introduction of triazole group in the polypeptide side chain exhibits both proton donor and accepting capability, thereby disrupting the secondary alpha helix conformation whereas these polypeptides regain secondary conformation upon protonation. Using this strategy, they have designed cell-penetrating polypeptides that have membrane penetration capability. This is a clever way of modulating secondary structures in polypeptides. While there are a lot of amide isosteres available, why did the authors choose only triazole and did the authors also attempt to use other isosteres? The manuscript and supporting information are well documented. It will be great if the authors could comment on this. I recommend publication of this article.

Reviewer #3 (Remarks to the Author):

This manuscript describes a facile methodology to control the conformation/secondary structure of polypeptides using donor-acceptor interactions. 1,2,3-triazole groups were conjugated onto the polypeptide side chains through "click" chemistry. The side-chain H-bonding ligands can serve as both H-bond donors and acceptors at neutral pH but only act as donors at protonated condition in acid solution, thereby to switch off/on the α -helical structure of polypeptides. The work presents a very elegant design of the stimuli-responsive polypeptides and exemplifies rational combination of the fundamental aspects of chemistry and materials characterization with biomedical application. The authors have clearly illustrated the mechanism behind off/on of α -helical structure, aroused from donor-acceptor pattern alternation of triazole side chains. The manuscript is also well-written. The reviewer would highly suggest its publishing after some minor revision.

1. The author nicely synthesized a series of polypeptides (P1 to P5). Only P2 has side chains with ethyl ammonium and all others are methyl ammonium. Is there any particular reason for this?

2. In Figure 1e, the authors demonstrated the spacing between triazole group and polymer backbone can strongly influence the resulting conformation. When triazole groups are placed far away from the backbone, P3 would retain α -helical conformation. It is a bit strange that the intensity of P3 is much lower than P2. The authors should provide some explanations.

RESPONSES TO REFEREES:

(Reviewer comments in black, our response in blue, and text added to the paper in magenta)

Reviewer 1

Remarks to the Author

This is a clever and interesting paper that deserves publication in Nature Materials. The synthetic chemistry is original and effective and the end result of switchable hydrogen bonding can have many useful follow-ons. I think the paper is very clear, except on one point. That is in the application to cell-penetration. I am completely convinced of the cell penetration observed in Figure 5. I am much less convinced of the evidence for endosomal escape. All of the fluorescence in Figure 5 appears punctate to me, indicating compartmentalization in endosomes. If this point is clarified, I am very enthusiastic about this paper.

We appreciate your positive comments.

The polypeptides at neutral pH adopted random-coiled structure and they were internalized via folate receptor-mediated endocytosis, as evidenced by the notably inhibited cellular uptake level at 4 °C compared to that at 37 °C. This new uptake data was incorporated as Supplementary Figure 10 (Page S26, highlighted). As such, we observed punctated spots in the CLSM image of HeLa cells following incubation with **P1(L)**-FA-FITC and **P1(DL)**-FA-FITC for 4 h, a typical phenotype of endocytic vesicles (endosomes) (Figure 5d, e). However, the quantitative analysis revealed that the co-localization ratio between **P1(L)**-FA-FITC and LysoTracker-Red-stained endosomes (40.5%) was remarkably lower than **P1(DL)**-FA-FITC (79.5%), which indicated that the conformational transition of **P1(L)**-FA-FITC to the α -helical state in endosomes promoted the endosomal escape of the polypeptide. Since the incubation time was relatively low (4 h), the escaped polypeptide may still stay close to the original endosomes instead of getting distributed to the whole cytoplasm. As such, we did not observe the aggregation/smear of fluorescence in the CLSM image.

To support such assumption, we incubated the cells with **P1(L)**-FA-FITC for prolonged time (8 h) before CLSM observation. It could be noted that large quantities of green dots fused together and distributed to large areas in the cytoplasm, and some of them appeared permeation patterns as expected, which indicated the effective escape of **P1(L)**-FA-FITC. In consistence with such finding, the colocalization ratio between **P1(L)**-FA-FITC and LysoTracker-Red-stained endosomes further decreased to 26.9%. The new CLSM images were added as Supplementary Figure 12 (Page S28, highlighted).

Reviewer 2

Remarks to the Author

The authors have developed a new strategy to modulate the conformation of polypeptides via proton acceptor-donor interactions arising from the side-chain H-bonding ligands. They have clearly demonstrated that introduction of triazole group in the polypeptide side chain exhibits both proton donor and accepting capability, thereby disrupting the secondary alpha helix conformation whereas these polypeptides regain secondary conformation upon protonation. Using this strategy, they have designed cell-penetrating polypeptides that have membrane penetration capability. This is a clever way of modulating secondary structures in polypeptides. While there are a lot of amide isosteres available, why did the authors choose only triazole and did the authors also attempt to use other isosteres? The manuscript and supporting information are well documented. It will be great if the authors could comment on this. I recommend publication of this article.

We thank the reviewer for the positive comments.

We are aware of the existence of other amide isosteres, including thioamides, sulfonamide, trifluoroethylamine, azo compound, and other nitrogen-based heterocycles (*ChemBioChem*, **2011**, *12*, 1801). Our plan is to make a complete understanding of the side-chain H-bonding pattern effect using triazole polypeptides, before further extending to other isosteres. We have two reasons to select triazole over other isosteres: (1) the pH-dependent H-bonding pattern, (2) the easy access of a series of polypeptides with the efficient Huisgen click chemistry.

Incorporating other amide isosteres onto the side-chains of polypeptides is our next step plan and will be included in our future publications. For instance, we are currently designing thioamide and sulfonamide based polypeptides using L-lysine as the starting materials to study the impact of these two groups (both BHB pattern) on polypeptide conformation.

Reviewer 3

Remarks to the Author

This manuscript describes a facile methodology to control the conformation/secondary structure of polypeptides using donor-acceptor interactions. 1,2,3-triazole groups were conjugated onto the polypeptide side chains through “click” chemistry. The side-chain H-bonding ligands can serve as both H-bond donors and acceptors at neutral pH but only act as donors at protonated condition in acid solution, thereby to switch off/on the α -helical structure of polypeptides. The work presents a very elegant design of the stimuli-responsive polypeptides and exemplifies rational combination of the fundamental aspects of chemistry and materials characterization with biomedical application. The authors have clearly illustrated the mechanism behind off/on of α -helical structure, aroused from donor-acceptor pattern alternation of triazole side chains. The manuscript is also well-written.

We appreciate the positive comments from the reviewer.

The reviewer would highly suggest its publishing after some minor revision.

1. The author nicely synthesized a series of polypeptides (P1 to P5). Only P2 has side chains with ethyl ammonium and all others are methyl ammonium. Is there any particular reason for this?

Thank you for your comments. Polypeptides **P1** and **P3-P5** were synthesized via click chemistry between polypeptide side-chain azides and ammonium-based alkynes, but **P2** was synthesized through the nucleophilic reaction between polypeptide side-chain chlorines and tertiary amines (Supplementary Scheme 3 and 4). For the synthesis of **P2**, the ammonium based polypeptide without side-chain triazoles, we actually first tried to use trimethylamine as the nucleophile. However, the reaction gave us a messy NMR spectrum and some water-insoluble polymeric residues after dialysis. We attributed the side reactions to the high reactivity of trimethylamine, and therefore used triethylamine as the nucleophile instead. The triethylamine reaction gave us a very clean NMR spectrum and polypeptides with very good water solubility.

We have synthesized a series of ammonium based polypeptides before (similar structure with **P2** but with different ammonium substitutions), and the analysis of their conformation revealed that the ammonium substitutions do not significantly alter the secondary structure (data not shown). We therefore assume the trimethylammonium analog of **P2** should have very similar α -helical conformation with **P2**.

To clarify this issue, we incorporated a short paragraph in Supporting Information (Page S13, highlighted):

“We first tried to synthesize trimethylammonium based polypeptides for good comparison with **P1**, however, the reaction failed since the high reactivity of starting material trimethylamine caused serious side-reactions. We therefore use triethylamine instead as the nucleophile.”

2. In Figure 1e, the authors demonstrated the spacing between triazole group and polymer backbone can strongly influence the resulting conformation. When triazole groups are placed far away from the backbone, P3 would retain α -helical conformation. It is a bit strange that the intensity of P3 is much lower than P2. The authors should provide some explanations.

Thank you for your comments. The elongation of spacer length only weakens the disrupting effect from BHB triazole groups but not completely deactivate them. As a comparison, there is absolutely no “disruptive” groups in **P2**, which adopts an α -helical conformation with high helicity. Although the triazole groups are placed further away from the backbone in **P3** compared with **P1**, they can still interact with the backbone amides in **P3**. This is evidenced by the fact that the conformation of **P3** is pH-dependent, which is very different with **P2** (pH-independent, Supplementary Figure 3). The pH-dependence of secondary structure originates from the side-chain triazole of **P3**, in a similar manner of **P1**, where the protonation of triazole alters its H-bonding pattern and further induces the change in its conformation.

Supplementary Figure 3 was modified according to your comments to elucidate the low helicity of **P3** (Page S19, highlighted). The CD spectra of P3 at various pH values were shown in Supplementary Figure 3b. The text below was also incorporated in Supporting Information (Page S19, highlighted):

“The conformation of control polypeptide **P3**, on the other hand, is pH-dependent due to the side-chain triazoles. Although the longer distance between triazole and backbone weakens the disrupting effect of triazole in **P3**, it cannot completely block the disruption. This also explains why **P3** has a much lower helicity than **P2** at pH 7.0 (Figure 2e).

REVIEWERS' COMMENTS:

Reviewer #1 (Remarks to the Author):

I am satisfied by the authors' response to my review.

Reviewer #2 (Remarks to the Author):

The authors have addressed all of my comments. Ready for acceptance.

Reviewer #3 (Remarks to the Author):

The authors have satisfactorily addressed my comments, thus the work is publishable in its current form.